# Ethylene-Propylene-Methylene/Isoprene Rubber/SiO_2_ Nanocomposites with Enhanced Mechanical Performances and Deformation Recovery Ability by a Combination of Synchronously Vulcanizing and Nanoparticle Reinforcement

**DOI:** 10.3390/polym16192809

**Published:** 2024-10-03

**Authors:** Rongyan Hu, Ran Xiao, Xinxin Xia, Yonggang Shangguan, Qiang Zheng

**Affiliations:** 1MOE Key Laboratory of Macromolecule Synthesis and Functionalization, Department of Polymer Science and Engineering, Zhejiang University, Hangzhou 310027, China; 12229007@zju.edu.cn (R.H.); 2023310064@link.tyut.edu.cn (R.X.); 21629048@zju.edu.cn (X.X.);; 2Shanxi-Zheda Institute of Advanced Materials and Chemical Engineering, Taiyuan 030032, China

**Keywords:** compound vulcanizates, nanocomposites, rheological behavior

## Abstract

It is highly desired yet challenging to develop advanced elastomers with excellent mechanical properties, including high strength and toughness. In this work, strong and tough rubber/rubber compound vulcanizates were facilely prepared by blending ethylene-propylene-methylene (EPM) and isoprene rubber (IR) together with dicumyl peroxide (DCP) and subsequent vulcanization, since both EPM and IR can be vulcanized synchronously by DCP and the well-crosslinked structure of EPM/IR vulcanizate presented a stable phase separation state. By tuning their composition, EPM/IR vulcanizates could present remarkably improved mechanical strength and toughness, as well as excellent energy dissipation and deformation recovery abilities. Furthermore, EPM/IR/SiO_2_ nanocomposites with better properties were prepared by introducing silica nanoparticles into EPM/IR vulcanizates. It was found that the high toughness and strength of EPM/IR vulcanizates and EPM/IR/SiO_2_ nanocomposites mainly resulted from the combination of stretchability of EPM and strain hardening of IR. Their excellent energy dissipation and deformation recovery abilities were related to the macromolecular characteristics of EPM and IR, compatibility between EPM and IR, and their crosslinking dynamics.

## 1. Introduction 

Rubber materials are widely used in industrial manufacture and daily life, in applications such as tires, seals, and shock absorbers [1]. Since almost all raw rubbers are mechanically weak, the reinforcement process before practical application is indispensable. Generally, there are three main approaches to the mechanical modification of rubber materials, which are filler-filling reinforcement [2,3,4,5], vulcanization by covalent bond [6,7], and sacrificial bond reinforcement [8,9,10], respectively. Although these methods could remarkably improve the mechanical properties of certain rubber materials, they all have their own problems. For example, it is difficult to distribute nanoparticles homogeneously in the rubber matrix and to control the interface between filler and matrix for filling reinforcement [11,12,13]. In addition, the trade-off between the strength and toughness of rubber vulcanizated by covalent bonds still exists [14]. Furthermore, toughening by introducing sacrificial bonds into the rubber is only applicable to certain special elastomers, and usually involves complex chemical modifications [15,16,17,18]. Thus, considering the great application potential of elastomers in practical industry and plastic toughening, it is desirable to develop a facile method to prepare rubber materials with excellent mechanical properties. 

Polymer blending is an important way to mechanically modify polymeric materials and has also been applied in rubber–rubber systems [19,20,21,22,23]. Blending allows the complementary properties of various materials to enhance rubber’s service and processing performance. However, most of the material systems were based on natural rubber (NR) and other rubber materials were just incorporated into NR to improve its certain functionalities [24,25]. Synthetic isoprene rubber (IR) and NR are similar in structure and properties, and IR is widely used in the tire industry for its high tensile strength, good resilience, and good tack. IR has been used in blends with major polydienes like polystyrene butadiene rubbers (SBR) and polybutadiene (BR) rubbers to improve their processability and other mechanical properties [26,27]. However, as the compatibility between different rubber materials is usually so poor that the compatibilizers are required to prepare rubber/rubber composites, this would increase the cost to some extent and the complexity of industrial processing [28,29,30,31]. As a result, new methods for the concise and efficient utilization of synthetic rubber blends remain an important aspect of research and application.

In recent years, compound rubber has been extensively utilized to enhance the toughness of plastics, owing to its ease of processing and customizable properties [32,33]. In polypropylene (PP) impact modification, EPM and ethylene-propylene-diene monomer elastomer (EPDM) are the two most widely used elastomers. EPM exhibits high resistance to low temperatures and solvents and has a broad range of viscosity variations, offering excellent fluidity and plasticity during processing. Therefore, EPM blends have been extensively used to enhance toughness and improve performance in low-temperature applications of plastics [34,35,36]. Jia et al. employed a blend of styrene-[ethylene-(ethylene-propylene)]-styrene block copolymer (SEEPS) and EPM to improve the low-temperature toughness of PP, achieving a significant enhancement without compromising strength or stiffness [35]. Kim et al. dispersed EPM and poly(ethylene-octene) (EOC) within the PP matrix, achieving impact strengths exceeding 70 kJ/m^2^ even at −40 °C [36]. Zhao et al. carried out a comprehensive study on toughening polypropylene using EPM and SiO_2_ and proposed a toughening mechanism for composites [37]. These results indicate that the toughening efficiency of rubber in plastics is closely associated with the structure and properties of the compound rubber within the matrix, but the underlying toughening mechanism remains unclear, necessitating further investigation. Exploring high-performance compound rubber is crucial for advancing the application of rubber materials and the development of polymer toughening theory.

In this work, we present the fabrication of durable and tough EPM/IR compound vulcanizates, prepared through a simple process of blending EPM and IR with DCP and simultaneous vulcanization. Through detecting the morphologies, tensile, and shear behavior of the compound vulcanizates, the microstructure of the vulcanizates was carefully studied, and the origins of their excellent mechanical properties were discussed. In addition, by introducing silica nanoparticles and adjusting their distribution, EPM/IR nanocomposites with excellent mechanical properties were prepared, and their dynamic rheological behaviors were systematically studied.

## 2. Experimental Section

### 2.1. Materials

Ethylene propylene rubber (EPM, J-0050, with an ethylene mass fraction of about 45%, *M*_w_ = 1.85 × 10^5^, *M*_w_/*M*_n_ = 3.11) and isoprene rubber (IR, 2200, with a *cis* content of 98% or more) were commercial products obtained from Jilin Petrochemical (Jilin, China) and Duokang (Shanghai, China), respectively. Dicumyl peroxide (DCP) was purchased from Aladdin Chemistry (Shanghai, China). Fumed silica, Aerosil^®^ R974 (hydrophobic modification of A200 (hydrophilic, with a surface area of 200 m^2^/g with silanols partially replaced by methyls)) was purchased from Evonik Degussa (Parsippany, NJ, USA). The silanol density of R974 is 0.39 nm^−2^. All raw materials were not treated more before use.

### 2.2. Sample Preparation

EPM was compounded with IR as the designed mass ratio on an internal mixer HOOK (Kechuang Shanghai, China) at 110 °C for 10 min. The composites were prepared by introducing hydrophobic silica into the above EPM/IR compounds at 110 °C for 20 min. Then, the EPM/IR compounds and EPM/IR/SiO_2_ composites were hot-pressed at 140 °C and 14.5 MPa for 45 min on a vulcameter XL-25 (Xinli Rubber Machinery Huzhou, China), respectively. The vulcanized compounds and composites were crosslinked by adding DCP during the last 3 min of mixing in an internal mixer. The other processes were the same as those used for the compounding of the unvulcanized samples. For brevity, samples are represented as EPM 40 in some figures, which indicates that the mass fraction of EPM is 40% in the EPM/IR compounds. The rest can be completed in the same manner.

### 2.3. Characterization

#### 2.3.1. Tensile Test

Uniaxial tensile tests were conducted on an electronic universal testing machine Instron 3343 (Norwood, MA, USA) at room temperature. Testing was performed using the ASTM D412 standard, and all samples were cut from vulcanized rubber sheets into dumbbell shapes with an initial gauge length of 25 mm and a width of 4 mm. The tensile speed was 500 mm/min. The result was an average of five specimens.

#### 2.3.2. Equilibrium Swelling

The swelling degree was measured when samples reached swelling equilibrium in tetrahydrofuran (THF) at room temperature, and the gel content was obtained after the swollen gels were dried thoroughly in a vacuum oven at 55 °C.

The volume fraction of the rubber network in swollen gel (*V_r_*) was calculated according to the following equation [38]:$$V_{r}=\frac{\left(m_{2}-m_{0}\varphi \right)/\rho_{r}}{\left(m_{2}-m_{0}\varphi \right)/\rho_{r}+\left(m_{1}-m_{2}\right)/\rho_{s}}$$
where *m*_0_ is the weight of the sample before swelling; *m*_1_ and *m*_2_ are the weights of the swollen and dried sample, respectively; *φ* is the weight fraction of the insoluble component; and *ρ_r_* and *ρ_s_* are the density of rubber and THF, respectively. The swelling ratio is defined as 1/*V_r_*.

The elastically active network density was calculated by the well-known Flory–Rehner equation:$$V_{e}=-\frac{\ln(1-V_{r})+V_{r}+\chi {V_{r}}^{2}}{V_{s}({V_{r}}^{1/3}-V_{r}/2)}$$
where *χ* is the Flory–Huggins polymer-solvent interaction parameter (0.397 for EPM and THF, 0.263 for IR and THF, and the parameter for the compound sample was calculated according to the linear addition of composition [39]), and *V_s_* is the molar volume of the solvent (81 mL/mol for THF at room temperature).

#### 2.3.3. Phase Contrast Microscopy (PCM)

PCM observations were conducted using an optical microscope (XSZ-HX, COIC, Chongqing, China) and the specimens were sandwiched between two microscope cover lips. 

#### 2.3.4. Transmission Electron Microscopy (TEM)

TEM (JEM-1230, JEOL, Tokyo Japan) was used to study the phase separation at an acceleration voltage of 100 kV. The double bond was stained with osmium, and then the samples were made to a thickness of about 100 nm by freezing and ultrathin slicing.

#### 2.3.5. Temperature Modulated Differential Scanning Calorimetry (TMDSC)

TMDSC tests were conducted on a differential scanning calorimeter (Q100, TA, Delaware, NC, USA). Before measurement, a calibration for determining the accurate heat capacity of sapphire was performed at a heating rate of 1 °C/min, with a modulating amplitude of 1 °C and a period of 120 s. Samples of 5–10 mg sealed in aluminum pans were cooled to −90 °C at 1 °C/min. After equilibrating at −90 °C for 5 min, the samples were heated to −45 °C at 1 °C/min and their heat capacity was recorded. The period and amplitude of the modulated signal were set to 120 s and ±1 °C, respectively.

#### 2.3.6. Dynamic Mechanical Analysis (DMA)

Dynamic mechanical analysis was performed on a dynamic mechanical analyzer (Q800, TA, Delaware, NC, USA). A double cantilever mode with a displacement of 100 μm was adapted. The samples were scanned from −80 to 20 °C under a frequency of 10 Hz and a heating rate of 3 °C/min.

#### 2.3.7. Rheological Measurement

The rheological measurements were conducted on an advanced rheological expansion system (ARES-G2, TA, Delaware, NC, USA) with a 25 mm parallel plate setup. Strain sweeps from 0.01 to 100% were conducted at 40 °C with a constant angular frequency of 10 rad/s. Frequency sweeps were carried out from 100 to 0.05 rad/s with a constant strain of 0.05% at different temperatures. Temperature sweeps from 40 to 160 °C were conducted with an angular frequency of 10 rad/s and a strain of 0.3%. Time sweeps were performed at 120 °C with an angular frequency of 10 rad/s and a strain of 0.3%. For the rheological tests, disk-like samples with a diameter of 25 mm and thickness of 2 mm were used.

## 3. Results and Discussion

### 3.1. Mechanical Properties

First, the mechanical properties of the two elastomers were examined. The interactions between adjacent EPM macromolecular chains are weak and flexible, resulting in minimal strain-hardening or strain-induced crystallization under large strains. Consequently, EPM exhibits relatively low mechanical strength. However, after vulcanization with dicumyl peroxide (DCP), the tensile strength of the EPM sample improved. Notably, different from tensile strength, the increase in stretchability is more pronounced with the addition of a specific amount of crosslinking agent, as shown in Figure 1a. In contrast, IR is a highly unsaturated diene rubber with weak polarity and excellent elasticity. As shown in Figure 1b, IR vulcanizates exhibit a distinct strain-hardening phenomenon as tensile strain increases, similar to the strain-induced crystallization behavior reported in previous studies on IR [40,41,42]. With the increase in the amount of DCP, the breaking strain generally decreases and the strength increases. However, excessive crosslinking density renders the material brittle, leading to a decrease in strength at higher DCP content. Therefore, IR has the best mechanical properties when crosslinked with 2 phr DCP.

Considering the facts that both EPM and IR could be vulcanized by DCP, combining the excellent stretchability of EPM vulcanizates with the notable strain-hardening characteristics of IR vulcanizates could result in compound vulcanizates with significantly enhanced mechanical properties compared with pure EPM or IR vulcanizates. Figure 1c shows the schematic for the preparation of EPM/IR compound vulcanizates. It can be seen from Figure 1a,b that the optimal content of DCP for crosslinking pure EPM or IR is 2 phr, therefore the DCP usage of 2 phr is adopted for EPM/IR compound vulcanizates in the following experiments.

Figure 2 gives the tensile results of EPM/IR and EPM/IR/SiO_2_ vulcanizates with various compositions. Tensile parameters are shown in Appendix A. The results show that EPM/IR vulcanizates with specific EPM contents demonstrate significantly enhanced mechanical properties. As shown in Figure 2a,b, the breaking strength and toughness of EPM/IR vulcanizates increase dramatically compared with pure EPM or IR vulcanizates when the EPM content is about 20%~40%. Additionally, the elongation at break for EPM/IR vulcanizates with EPM contents of 20%, 30%, or 40% is nearly identical to that of pure EPM vulcanizates. In addition, it can be seen that the Young’s modulus of EPM/IR vulcanizates increases almost linearly with the increase in EPM content, indicating a relatively homogeneous distribution of EPM phases in EPM/IR vulcanizates. 

To investigate the effect of crosslinking density on the mechanical properties of EPM/IR vulcanizates, tensile tests for EPM/IR samples with varying amounts of DCP were conducted. The results, provided in Appendix A, show that all EPM/IR vulcanizates exhibit excellent mechanical properties. However, the samples with 2 phr DCP demonstrate relatively superior overall mechanical performance, consistent with the behavior of pure EPM and IR vulcanizates, as shown in Figure 1.

As expected previously, the EPM/IR vulcanizates with 20, 30, and 40% EPM content show significantly enhanced breaking strength and toughness, indicating the effective combination and complementarity of the advantages of the two components. When subjected to relatively small strain, network chains of EPM components in EPM/IR vulcanizates are easier to slide than those of IR, endowing the vulcanized sample with high stretchability and maintaining the integrity of the IR network. As the strain increases to a certain extent, the IR network in EPM/IR vulcanizates begins to play the role of strain-hardening, and subsequently the stress increases rapidly with the strain. In consequence, the EPM/IR vulcanizates exhibit high elongation ratios and breaking strength simultaneously. 

Figure 2c,d shows tensile properties of EPM/IR/silica vulcanizates with varying silica content. The incorporation of silica significantly enhances the modulus and breaking strength of EPM/IR composites. However, the elongation at break decreases as silica content increases. Interestingly, the effect of silica loading on toughness is more nuanced: toughness improves slightly at low silica content but decreases markedly at higher silica levels. When the silica content reaches 40 phr, the sample becomes too brittle, and its mechanical properties deteriorate significantly. These findings suggest that the mechanical properties of EPM/IR/silica composites can be quantitatively adjusted by changing the silica content.

### 3.2. Hysteresis and Recovery Ability

In order to understand the viscoelastic behavior of the EPM/IR/SiO_2_ composite, we first investigated the hysteresis of sequential cyclic loading–unloading curves of EPM/IR vulcanizates, stretched to different maximum applied strain, and their deformation recovery ability (as shown in Figure 3). This analysis helps elucidate the contribution and mechanism of the EPM and IR networks to the superior mechanical properties of these composites [43,44]. Notably, significant hysteresis is observed during the initial loading–unloading cycles of pure EPM vulcanizates, indicating substantial energy dissipation due to the motion and friction of EPM chains under applied stress. As the IR content increases, the stress at a given maximum strain increases, and the area of hysteresis cycles decreases, indicating improved elasticity in the EPM/IR vulcanizates. Additionally, the overlap between the second and first loading–unloading curves, measured 30 min apart, initially increases (as shown in Figure 3a–d) and then decreases gradually (as shown in Figure 3e,f) as the IR content increases. This suggests that the recovery ability of the compound rubbers depends on their composition and can be qualitatively adjusted by composition. Notably, the second loading curve of the sample with 40% EPM nearly coincides with the first, further confirming that the compound rubber with 40% EPM exhibits superior mechanical properties.

Figure 4 gives the corresponding quantitative energy dissipation results for various EPM/IR vulcanizates during the first and the second loading, respectively. In general, the energy dissipation value represents the hysteresis, while the energy dissipation ratio of the sequential two deformations indicates the resilience or elastic stability of the structure. A smaller hysteresis of pure EPM vulcanizates during the second loading is observed because the energy dissipation value of the second loading is much lower than that of the first loading, and the EPM sample even breaks before the completion of the second loading as shown in Figure 3a. With the increase in IR content, the deformation recovery ability of EPM/IR vulcanizates significantly improves, and the energy dissipation ratios for EPM/IR vulcanizates with EPM of 20%, 40%, 60%, and 80% could all remain over 80% at various maximum strains, which are even higher than pure IR vulcanizate, suggesting that the IR component in EPM/IR vulcanizates could indeed play a role in improving the deformation recovery ability of EPM. It should be pointed out that the recovery ability of EPM/IR vulcanizates is better than that of pure IR vulcanizates, except under small strains. The reason for this may lie in two factors: (1) under small strains, the IR network can sustain greater stress, thereby reducing hysteresis caused by the motion of EPM molecules; (2) under large strain, the high stretchability of the EPM network may protect the integrity of the IR network through the sliding of EPM chains, and the elasticity of the intact IR network can promote the deformation recovery of EPM component. Thus, EPM/IR vulcanizates with certain compositions can exhibit both the energy dissipation ability of EPM and the elasticity of IR, which together contribute to their enhanced mechanical properties.

To investigate the dissipation and the recovery effect of nanoparticles on EPM/IR vulcanizates, the hysteresis of sequential cyclic loading–unloading curves for various EPM/IR/SiO_2_ vulcanizates with different silica loadings are shown in Figure 5. For the EPM/IR vulcanizate with 40% EPM content, as the silica loading increases, the stress at a given maximum strain rises, and the hysteresis area expands, indicating greater energy loss in the composites due to the presence of nanoparticles. However, when silica loading reaches 40 phr, the tensile stress decreases instead. Furthermore, the overlapping degree between the second loading–unloading curves and the first loading–unloading curves decrease gradually (as shown in Figure 5b–d) with the silica content, indicating a reduced deformation recovery ability induced by silica nanoparticles. These results show that the silica loading should not be too high when reinforcing EPM/IR vulcanizate with silica nanoparticles, as excessive nanoparticles can hinder the recovery of rubber materials after large deformations.

### 3.3. Crosslinking Structure and Microstructure

Crosslinking plays a crucial role in determining the mechanical properties of rubber materials. The vulcanization carried out in our system is a radical-initiated process in which primary radicals produced by the thermal decomposition of peroxides cause the addition of double bonds, the abstraction of hydrogen atoms, and the coupling of two radicals. As shown in Appendix A, pure IR is more quickly crosslinked by DCP compared with pure EPM due to its unsaturation. Thus, in EPM/IR and EPM/IR/silica vulcanizate samples, whether EPM and IR could be simultaneously crosslinked by DCP or whether DCP would aggregate mainly in only one rubber still needs to be probed. Swelling equilibrium is the most common way to characterize the crosslinking structure of vulcanized rubber materials, as shown in Figure 6. It can be seen that the swelling ratio increases, and elastic active network chain density decreases with the increase in EPM content. We can see that IR has a higher crosslink density due to the higher activity of its double bonds. The amount of crosslinked insoluble products is larger than the amount of IR containing double bonds in EPM/IR, thus indicating that also saturated EPM chains are submitted to the crosslinking process. Considering that peroxide-derived radicals react promptly with surrounding molecules, the addition of double bonds is in competition with the less favored hydrogen abstraction from the saturated EPM phase and may lead to a well-crosslinked IR with some grafting EPM in the final blends [45,46]. The similar structures of crosslinked IR with EPM and some EPM grafting promote the compatibility of the two phases, resulting in blends with excellent mechanical properties. For the EPM/IR/silica vulcanizate samples, a high content of silica will accelerate the vulcanization process, as shown in Appendix A.

Most polymer blends are immiscible for lack of strong intermolecular interactions, and almost all the polymer blends in practical application are phase-separated systems [47,48]. For these polymer blends, the compatibility between different components, the interfacial strength, and the phase morphology are important factors influencing the mechanical properties of the products [24,29]. Here, PCM and TEM were used to observe the phase morphology of the EPM/IR vulcanizates. As shown in Figure 7a–d, apparent phase separation between the EPM and IR components can be observed in the EPM/IR vulcanizates with compositions of 20/80 (island structure) and 40/60 (bi-continuous structure) by weight, which exhibit comprehensively improved mechanical properties, as shown in Figure 1. It is worth noting that although these PCM images are from the unvulcanized EPM/IR samples, the morphological results of these unvulcanized rubber can theoretically reflect the phase structure of crosslinked samples because the crosslinking occurs in situ. The size of phase domains ranges from several to dozens of microns, which is also confirmed by the TEM results of osmium-stained EPM/IR vulcanized samples, as shown in Figure 7c,d. For EPM/IR/silica vulcanizates, the addition of silica does not affect the two-phase separation, as shown in Appendix A. Both the nanoparticles and the osmium-stained IR showed a darker color, so it is difficult to observe the distribution of nanoparticles from TEM images. However, there is no silica in the uncolored EPM phase, indicating that silica is mainly distributed in IR due to the great differences between the polar character of the silica surface and the non-polar EPM rubber matrix.

As an efficient way to explore the interface interactions between diverse phases of polymer blends, temperature-modulated differential scanning calorimetry (TMDSC) can detect changes in glass transition temperature (*T*_g_) and reversible heat capacity (*C*_p_) of components in blends to characterize the interfacial strength [49,50,51]. As shown in Figure 8a,b, *T*_g_ of unvulcanized pure EPM and IR are −62 °C and −66 °C, respectively, which are so close that there is almost only one glass transition step on the *C*_p_ ~ temperature curves of various EPM/IR compounds. Similar *T*_g_ results for EPM/IR compounds were also obtained from DMA measurements (as shown in Appendix A). Previous studies have shown that a single *T*_g_ appearing in the blend is not enough to indicate that the blends are compatible when the *T*_g_ difference between two components in blends is less than 20 °C [52]. However, the EPM/IR blends exhibit a single *T*_g_ which changes almost linearly with the composition, indicating that the two components are similar in structure to some extent. In addition, the increment of reversible heat capacity (*ΔC*_p_) at the glass transition region of compound samples and pure rubber samples usually reflects the internal interfaces between the diverse components [53]. Compared with the ideal linear addition of *ΔC*_p_ of EPM/IR compound rubber as indicated by the blue dashed line in Figure 8c, there is no obvious positive deviation or negative deviation for the *ΔC*_p_ of various EPM/IR compound samples, suggesting that the interfacial interaction between EPM and IR phases is neither strong nor too weak [54]. Thus, both the *T*_g_ and *ΔC*_p_ results suggest that the intermolecular interaction between EPM and IR may be similar to that in pure EPM or pure IR. However, due to the self-connectivity of component molecules, the unvulcanizated EPM/IR compound is still in a phase-separated state but with proper phase morphology, as confirmed by Figure 7.

For thermal stability, the thermogravimetric analysis curves of different EPM/IR/silica vulcanizates are shown in Appendix A. The weight loss temperature of 340 °C in the first stage is due to the thermal decomposition of IR, and the temperature of 400 °C in the second stage is due to EPM.

### 3.4. Dynamic Rheological Behavior

Figure 9 demonstrates strain sweep curves and frequency sweep curves of various EPM/IR vulcanizates. Similar to the Young modulus results of pure EPM and IR vulcanizate in Figure 2, the storage modulus (*G*′) of pure EPM vulcanizate under small strain is also larger than that of pure IR, and the moduli of EPM/IR vulcanizates increase significantly with the increase in EPM content. In addition, with the increase in strain, there is a critical strain for the modulus of all vulcanizates, and beyond this critical strain these vulcanizates present nonlinear rheological behavior, i.e., the storage modulus begins to decrease while the loss modulus presents an overshoot peak, which is similar to the results reported in previous studies [12,13,55]. It is noted that *G*′ of pure EPM vulcanizate decreases rapidly while that of pure IR vulcanizate decreases much less when the strain is higher than the critical strain. For the EPM/IR vulcanizates, the *G*′ decrease in EPM/IR (20/80) vulcanizate, due to the higher IR content, is close to the case of pure IR vulcanizates, while that of the EPM/IR (40/60) vulcanizate is located between the EPM/IR (20/80) vulcanizate and pure EPM vulcanizate. These results may be ascribed to the strain-hardening properties of IR vulcanizates.

The frequency sweep results of EPM/IR vulcanizates show an interesting phenomenon. The *G*′~*ω* curve of pure EPM vulcanizate presents an obvious frequency dependence, while the *G*′ of pure IR and EPM/IR vulcanizates with less EPM, and hardly changes with frequency. It is suggested that not only the IR component is more easily crosslinked by DCP, as mentioned above, but also that the crosslinked network of the IR component is more perfect than that of EPM. This result can also be further verified by loss modulus (*G*″). Although the *G*′~*ω* curves of pure IR and EPM/IR vulcanizates with less EPM are very close, the difference in their *G*″~*ω* is very obvious. *G*″ of EPM/IR vulcanizates increases significantly with the increase in EPM content, indicating that there is more internal friction in the small deformation process due to an EPM network with an imperfect crosslinking structure.

For EPM/IR/silica vulcanizates, their rheological behavior is significantly different from those of EPM/IR vulcanizates due to the introduction of nanoparticles. Figure 10 gives strain sweep and frequency sweep results of different vulcanizates, respectively. It can be seen from Figure 10a that the *G*′ of EPM/IR/silica vulcanizates is significantly higher than that of EPM/IR vulcanizate, and is close to that of EPM/silica vulcanizate. Furthermore, the EPM/IR/silica vulcanizates present a larger critical strain and a slower decline in modulus than the EPM/silica vulcanizate, indicating that silica nanoparticles have a better reinforcing effect on the EPM/IR compound matrix than on the pure IR or EPM matrix.

Figure 10b gives *G*″ for various vulcanizates during the strain sweep process. It can be found that the *G*″ of EPM/IR/silica vulcanizate is closer to that of EPM/silica vulcanizate, but its overshoot peak appears at the higher strain due to the existence of nanoparticles. At the same time, because of the presence of nanoparticles, the *G*″ of EPM/IR/silica vulcanizate is much larger than that of EPM/IR vulcanizate without nanoparticles. Taking the *G*′ results of various vulcanizates in Figure 10a into account, it seems that the presence of silica nanoparticles plays a role in counteracting the modulus decrease caused by the presence of IR rubber. Furthermore, the frequency sweep results of different vulcanizates also seem to confirm the above point: the *G*′ curve of EPM/IR/silica vulcanizate almost coincides with that of EPM/silica vulcanizate but is significantly higher than that of EPM/IR. At high temperatures, these vulcanizates show similar results, as shown in Appendix A. The rheological results of EPM/IR/silica with different silica loadings are shown in Appendix A. The *G*′ and *G*″ of the composites increase with the increase in silica loading because of the reinforcement effect of nanoparticles. The particle-macromolecular structure formed in the composites is more easily destroyed than the macromolecular structure with the increase in the silica content, resulting in a decrease in the critical strain. 

The phase-separated structure of EPM/IR compound rubber has been proved from the above morphology investigation, TMDSC, and rheological measurements. In the practical application of polymer blends, the stability of the phase structure would also have an important impact on the performance of the products. The time–temperature superposition (TTS) principle is a common method used to characterize the much broader time interval [56], and it also can reflect the phase structure stability of the blending system [23]. As shown in Figure 11, smooth superposition master curves at the reference temperature of 40 °C could be obtained, demonstrating that the EPM/IR compound vulcanizates with various composition ratios all follow the TTS principle well, and suggesting that the phase structure of the EPM/IR phase is relatively stable within the testing temperature range. Stronger relaxations are to be expected for crosslinked EPM networks, which are probably bound up with their relatively high entanglement plateau modulus, high flexibility, low molar volume, and small chain thickness [57]. The addition of IR makes the compound have a flatter *G*′ curve, lower *G*″, and lower tan*δ*.

## 4. Conclusions

In conclusion, we have prepared mechanically tough and strong EPM/IR compound vulcanizates and EPM/IR/silica nanocomposites by synchronously vulcanizing EPM and IR with DCP and introducing silica nanoparticles. By tuning the composition of EPM/IR, the mechanical strength and toughness of EPM/IR compound vulcanizates could become 4~5 times higher than pure EPM and IR due to the combination of EPM’s stretchability and IR’s strain-hardening properties. The network chains of the EPM component in EPM/IR vulcanizates are easier to slide than those of IR, endowing high stretchability and maintaining the integrity of the IR network. Then, as the strain increases, the IR network begins to play a strain-hardening role, and subsequently the stress increases rapidly with the strain. Meanwhile, the EPM/IR compound vulcanizates could also possess outstanding energy dissipation and deformation recovery abilities simultaneously. Phase morphology investigations of EPM/IR compound rubber showed that this was a typical phase-separated system, and the island structure and bi-continuous structure can convert by changing their composition. Linear and nonlinear rheology of EPM/IR compound vulcanizates coincided well with that of the composition, due to the phase-separated structure of the compound sample, but the addition of nano-particle silica fillers compensates for the rheological changes caused by IR. Such tough and strong compound vulcanizates, with their energy dissipation and deformation recovery abilities, should have a wide spectrum of applications, such as in tough shock absorbers, elastic seal devices, etc. 

## Figures and Tables

**Figure 1 polymers-16-02809-f001:**
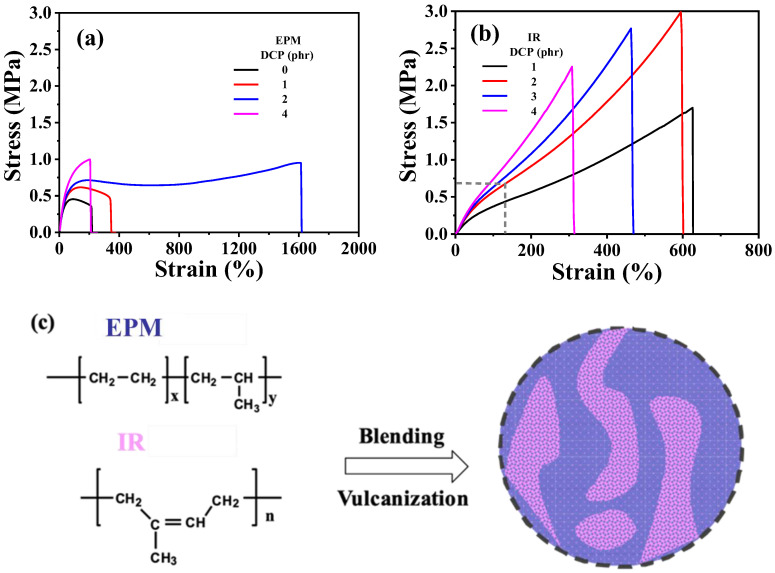
(**a**,**b**) Tensile stress–strain curves of vulcanized EPM and IR with different DCP content, respectively. (**c**) Schematic for sample preparation procedure of EPM/IR compound vulcanizates.

**Figure 2 polymers-16-02809-f002:**
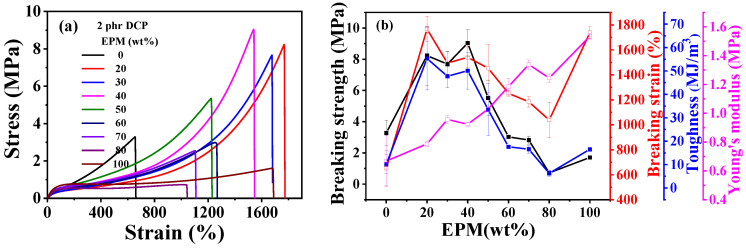
(**a**) Tensile stress–strain curves and (**b**) related mechanical properties of EPM/IR vulcanizates with various compositions. (**c**) Tensile stress–strain curves and (**d**) related mechanical properties of vulcanized EPM/IR/SiO_2_ composites with different silica loading.

**Figure 3 polymers-16-02809-f003:**
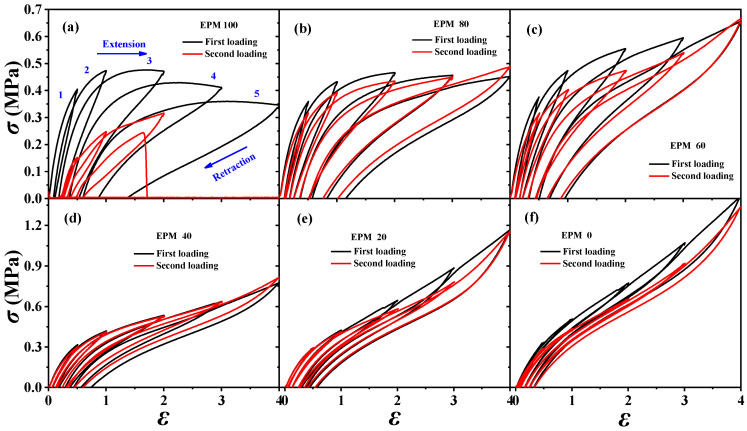
(**a**–**f**) Stress–strain curves of various EPM/IR vulcanizates during the first (black line) and second (red line) sequential loading–unloading tensile tests at a series of different maximum applied strains. The second loading is conducted 30 min after the first loading–unloading.

**Figure 4 polymers-16-02809-f004:**
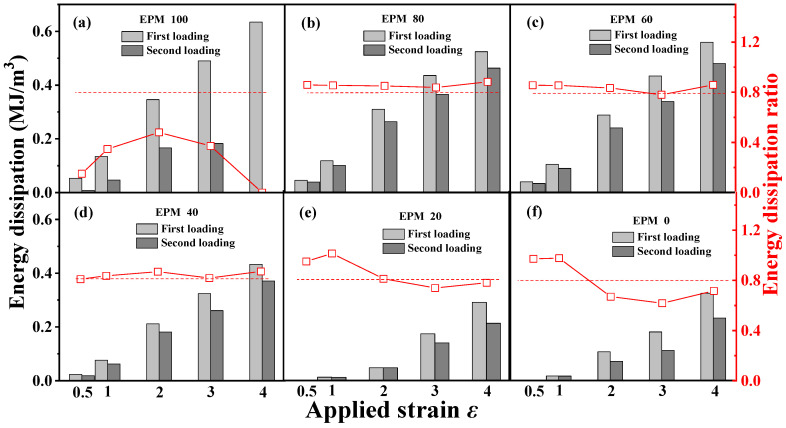
(**a**–**f**) Energy dissipation of cyclic tensile as a function of maximum strain for various EPM/IR vulcanizates during the first loading and second loading, and the corresponding energy dissipation ratio of the second loading to first loading. The second loading is conducted 30 min after the first loading–unloading.

**Figure 5 polymers-16-02809-f005:**
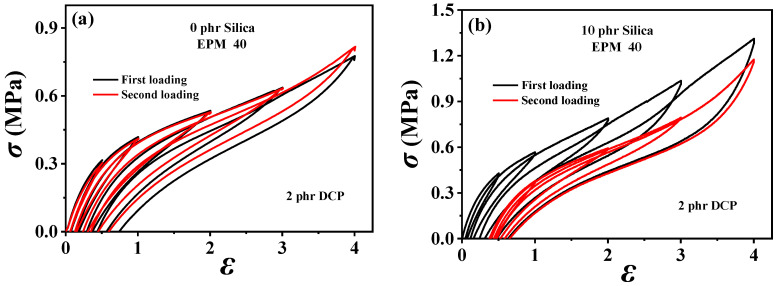
(**a**–**d**) Sequential loading–unloading tensile curves at a series of different maximum applied strains of various vulcanized EPM/IR/SiO_2_ composites with different SiO_2_ content (in the first loading (black line) and second loading 30 min after the first loading (red line)).

**Figure 6 polymers-16-02809-f006:**
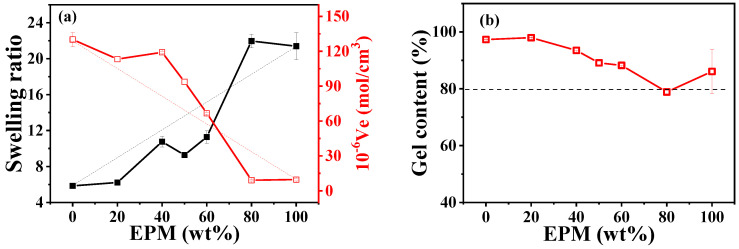
(**a**) Swelling ratio, elastic active chain density, and (**b**) gel content of various EPM/IR vulcanizates with 2 phr DCP.

**Figure 7 polymers-16-02809-f007:**
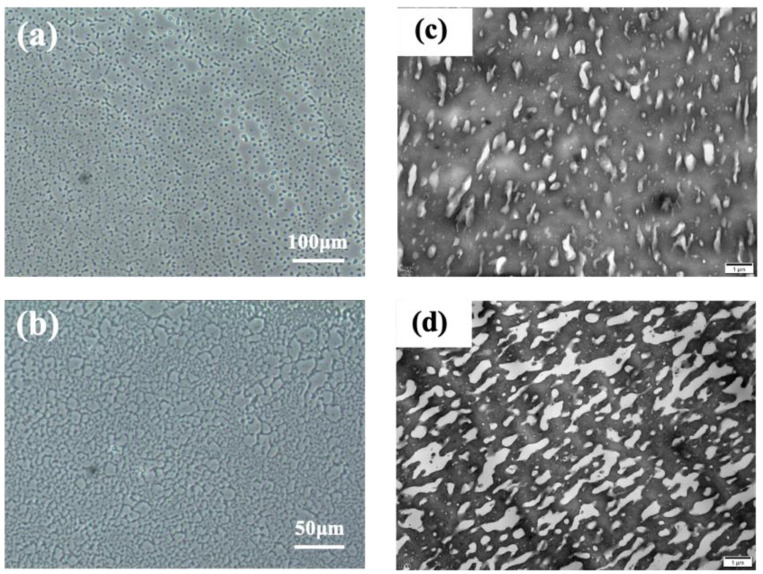
(**a**,**b**) PCM micrographs of EPM/IR unvulcanizate with compositions of 20/80 and 40/60; the black dots indicate the impurities on the PCM instrument lens. (**c**,**d**) TEM micrographs of EPM/IR vulcanizate with compositions of 20/80 and 40/60; the darker area is osmium-stained IR.

**Figure 8 polymers-16-02809-f008:**
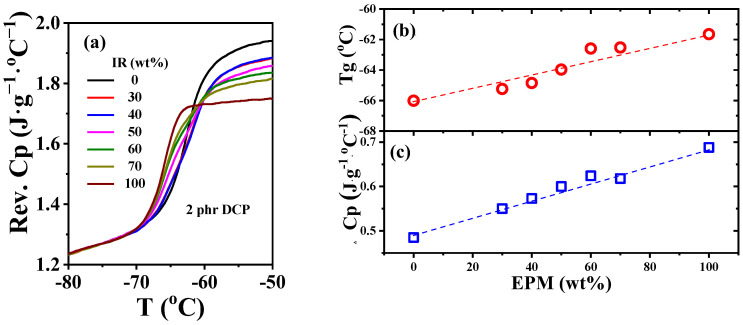
(**a**) Normalized specific reversible heat capacity curves and (**b**) glass transition temperature *T*_g_ and (**c**) heat capacity increment at *T*_g_ for EPM/IR compound rubber with different compositions.

**Figure 9 polymers-16-02809-f009:**
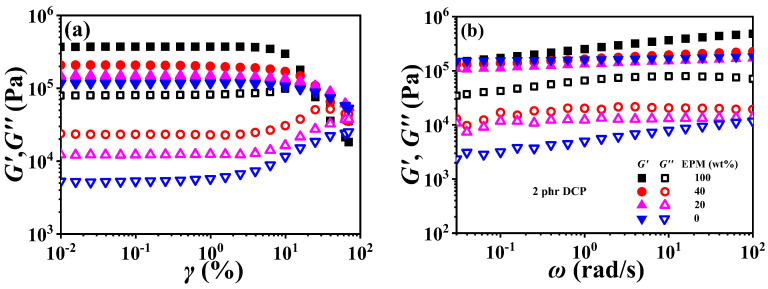
(**a**) Strain sweep curves and (**b**) frequency sweep curves of various EPM/IR vulcanizates.

**Figure 10 polymers-16-02809-f010:**
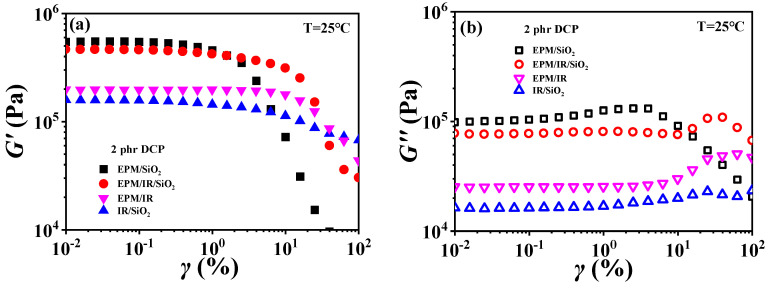
(**a**,**b**) Strain sweep curves and (**c**,**d**) frequency sweep curves of EPM/IR, EPM/silica, IR/silica, and EPM/IR/silica vulcanizates. Here, silica loading is fixed at 20 phr.

**Figure 11 polymers-16-02809-f011:**
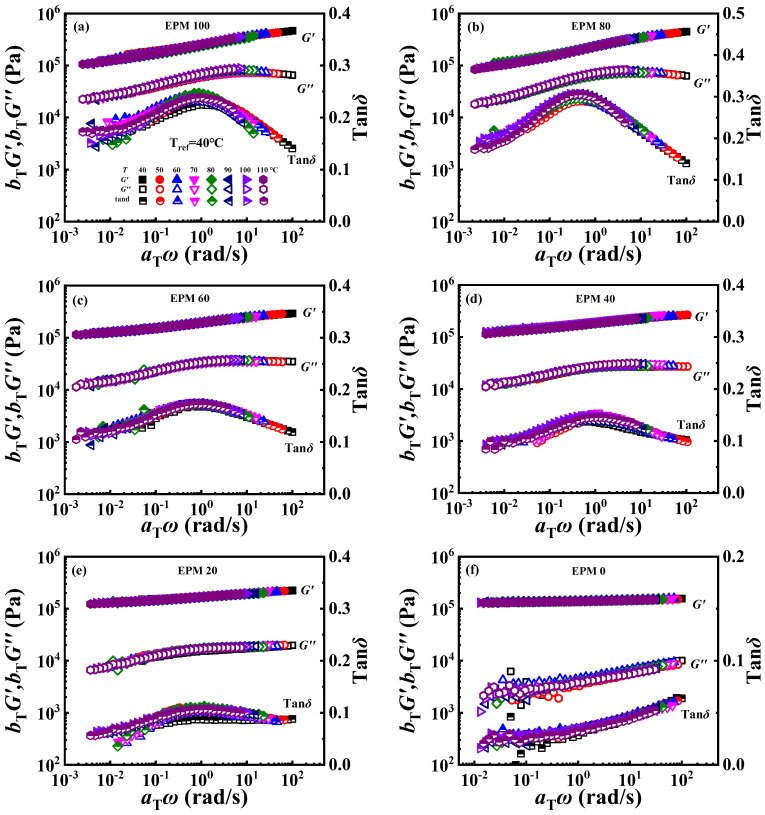
(**a**–**f**) Time–temperature superposition curves of various EPM/IR compound vulcanizates at reference temperature of 40 °C.

## Data Availability

Data will be made available on request. Data are contained within the article and Appendix A.

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
