# Peer review of "Ethylene-Propylene-Methylene/Isoprene Rubber/SiO2 Nanocomposites with Enhanced Mechanical Performances and Deformation Recovery Ability by a Combination of Synchronously Vulcanizing and Nanoparticle Reinforcement"

_polymers, 2024, doi:10.3390/polym16192809_

Round 1

Reviewer 1 Report

Comments and Suggestions for Authors

The paper deals with a well-known elastomer blends  studied by  a number of laboratories , also addition of nanosilica  and dynamic vulcanisation  are well known art. For these reasons the innovative  character of the paper is rather limited .Also results  analysis is very empirical and do not  goes inside the  mechanism of the free radical crosslinking in the two different elastomers.                                                                                    

The merit of the contribution consists of the extremely detailed and accurate  characterization of the samples prepared which  is certainly useful for researchers and mainly  technical operators in the field and the indication of a not innovative but well supported  procedure to modulate the compounds ultimate properties

Some points are indicated to help the decomposition  in this revision.

1.The authors suggest their procedure  as generally valid , but the results depends on the  starting material features. The two used polymer are commercial samples  and no data re reported about  their molecular weight , E/P ration in EPM , MWD, MW, cis content in IR ( synthetic and not NR !).Also the presence of aditives  should be considered as tey may interfere with the dynamic vulcanization process.

2. Presently the use of EPM acronym is suggested to replace EPR

3.As  also reported by the authors  oxygen free radicals produced by peroxide decomposition   are very reactive with the IR double bonds generating macromolecular radicals generating Carbon free radicals which can only give crosslinking with other free radicals but not extract  Hydrogen atom from EPM to generate the corresponding macromolecular radical. These last can only be produced by the primary oxygen radicals  , but this process is much less favored , from both thermodynamics and kinetic grounds, than attach to the IR double bonds .This would bring to the preliminary conclusion that  in the final blend IR is  well crosslinked  with some grafting with EPM favoring  the dispersion blending with the substantially not crosslinked EPM main part. This could explain  the observed mechanical behavior. The authors are then recommended to analyze these consideration and implement the paper consequently  from experimental and  speculative viewpoints. Some insights can be obtained by examining the paper : Ciardelli F, Dossi S, Galanti A, Magri A, Riolo S.  Molecular evolution during dynamic vulcanization of polyolefin mixtures for lead-free  thermoplastic vulcanized. Polym Adv Technol. 2019;1–9.

4.The English style needs  revision also to improve clarity of the discussion.

5-The introduction and the initial part of the discussion contain some well known information. They should be shortned and focused on the main objective of the paper

Comments on the Quality of English Language

Some revision  is recommended for better clarity

Reviewer 2 Report

Comments and Suggestions for Authors

Review Report _ polymers-3228390-peer-review-v1

I have reviewed the manuscript entitled “EPR/IR/SiO2 Nanocomposites with Enhanced Mechanical Performances and Deformation Recovery Ability by a Combination of Synchronously Vulcanizing and Nanoparticle Reinforcement’’. The authors need to address the following points before acceptance of the manuscript;

1.      Authors need to mention the benefits of using, particularly EPR and IR.

2.      In the introduction section, authors need to discuss and refer more work related to mechanical/toughening properties of polymer blends closely related to either EPR or IR, or both of them.

3.      Please mention the ASTM D standard for tensile test.

4.      In 2.3.2 section, authors need to put references for using the equations.

5.      What are the contrast modes in microscopy (PCM)? What is the advantage of phase-contrast microscopy over bright-field microscopy?

6.      In section 3.1, authors need to explain the reason behind this statement “With the increase of DCP amount, the breaking strain generally decreases but the strength first increases and then decreases’’.

7.      The authors should mention the tensile parameters like tensile strength, tensile modulus, and elongation at break within a table, and discuss accordingly.

8.      In 3.2 authors need to explain the reason behind this statement “As IR content increases, the stress at the same maximum strain increases and the area of hysteresis cycles becomes smaller, suggesting that the EPR/IR vulcanizates have much better elasticity’’.

Round 2

Reviewer 1 Report

Comments and Suggestions for Authors

Thanks for your  consideration of my comments and attempt to provide  some related improvement.

I still have  a few  remarks:

1.EPR was changed to EPM in the text but not in the Title

2. In the added text  on line 61 : PP and EPM  are certainly  partially compatible end miscible, but the  maromolecular strucrure is very different as even the used EPM haìs 45%  Ethylene content. Please  explain  in a more rigorous way.

3. My Comment 3 was aimed to  direct the authors towards  a  molecular explanation of the obtained properties  .The added sense:

". With the increase of DCP amount, the breaking strain generally decreases and the strength increases. However, excessive cross-linking density renders the material brittle,  leading to a decrease in strength at higher DCP content. Therefore, IR has the best mechanical properties when crosslinked with 2phr DCP"

is merely empiric . Please introduce the concepts related to the mechanism of the free radicals reactions with the two different polymers to provide a scientific  interpretation

Comments on the Quality of English Language

Some improvement was done. The text is in any case understandable.
